

# Current mean values in the XYZ model

Levente Pristyák[1,2] and Balázs Pozsgay[1,2*]

**1** Department of Theoretical Physics, Institute of Physics,
Budapest University of Technology and Economics, Műegyetem rkp. 3,
1111 Budapest, Hungary
**2** MTA-ELTE "Momentum" Integrable Quantum Dynamics Research Group,
Department of Theoretical Physics, Eötvös Loránd University,
Pázmány Péter stny. 1A, 1117 Budapest, Hungary

⋆ pozsgay.balazs@ttk.elte.hu

## Abstract

The XYZ model is an integrable spin chain which has an infinite set of conserved charges, but it lacks a global $U(1)$-symmetry. We consider the current operators, which describe the flow of the conserved quantities in this model. We derive an exact result for the current mean values, valid for any eigenstate in a finite volume with periodic boundary conditions. This result can serve as a basis for studying the transport properties of this model within Generalized Hydrodynamics.



# 1   Introduction

One dimensional integrable models are special many-body systems where analytical solutions are possible, and a number of physical quantities can be computed with exact methods [1, 2]. The study of these models goes back to the exact solution of the XXX Heisenberg spin chain by Hans Bethe [3]. The method of [3] is today known as the Bethe Ansatz, and together with its various generalizations [4] it is still the most general method to deal with integrable models.

The Bethe Ansatz is well suited for the computation of the finite volume spectrum of integrable models. Often it is also possible to study the thermodynamic limit, both for the ground state or even in finite temperature situations [5]. However, it is much more difficult to compute other physical quantities, like correlation functions and entanglement properties in equilibrium and out-of-equilibrium situations.

In this work, we consider special short-range correlation functions in the so-called XYZ model. More specifically, we investigate the finite volume mean values of those current operators, which describe the flow of the conserved quantities of the model. Our study is motivated by recent advances in two somewhat independent lines of research: Generalized Hydrodynamics on the one hand, and more traditional computations of exact correlation functions on the other hand.

Generalized Hydrodynamics (GHD) is a recent theory initiated in [6, 7] which aims to describe the non-equilibrium behaviour of integrable models. It incorporates the infinite number of local conservation laws characteristic for integrable systems, and also the closely related phenomenon of factorized and completely elastic scattering in these models. For a series of recent review articles we refer the reader to [8].

The first achievement of GHD was the exact treatment of ballistic transport in integrable models [6, 7, 9, 10], including the computation of the finite temperature Drude weights [11, 12][1] and the exact solution of certain selected quantum quench problems [16]. These computations are built on the local continuity equations which can be set up for the infinite family of local conserved charges. In these equations an essential ingredient is the dependence of the mean currents on the local equilibrium states. An exact formula for the mean currents was proposed in [6, 7] (and proven for relativistic QFT in [6]). This exact formula has a simple semi-classical interpretation: the currents are obtained as a summation over the charge values carried by the individual particles crossing a certain point, multiplied by an effective velocity which describes the propagation of wave packets in the background of the interacting particles [17].

For interacting spin chains the formula for the current mean values was proven in a series of works [18–20] with three different methods; the article [21] reviews these developments together with earlier work on the current mean values. The proofs of [18] and [20] were completely rigorous: they were built on a form factor expansion and on a new algebraic construction for the current operators. On the other hand, [19] pointed out connections between the current operators and long range deformed spin chains, but not all steps were mathematically rigorous. All three works focused on the case of the XXZ Heisenberg spin chain,

---

[1]For earlier computations of the Drude weights in the XXZ chain and the Hubbard model see [13–15].

but [19] contained a result also for the $SU(3)$-symmetric fundamental model. The construction of [20] is rather general: it applies to practically any Yang-Baxter integrable spin chain with local interaction. Nevertheless, a concrete application to a new situation has not yet been demonstrated.

In this work, we consider the current mean values in the XYZ spin chain, which is intimately connected with the 8 vertex model [1]. We work out the details of the construction of [20] and determine the exact formulas for the current mean values. The physical motivation to consider this specific model is the following. In the XYZ chain there is no $U(1)$-symmetry, thus there is no particle conservation in the Bethe Ansatz description. GHD itself does not rely on particle conservation on a fundamental level, nevertheless all previous cases were such that the collective excitations in a large system could be understood by the "dressing" of fundamental Bethe Ansatz quasi-particles. This idea underlies the semi-classical description of the theory, for example in the flea gas picture [17]. In contrast, there are no fundamental single particle states in the XYZ chain, and one needs to treat collective states even in a finite volume situation with small chain length. The formulation of GHD is in principle independent of the quasi-particle content: it is based on local continuity equations for the conserved charges. Therefore, it is interesting to work out the theory for the somewhat exotic case of the XYZ chain. As a first step, we start with the current mean values.

We should note that the transport properties of the XYZ model have already been investigated numerically in [22], where it was found that the clean model (without disorder) indeed shows ballistic transport, as expected from an integrable spin chain. We also note that the XYZ model has been experimentally realized with ultracold atoms [23].

The second main motivation for this work is more general: we wish to contribute to the theory of correlation functions in integrable spin chains. Exact computations for one-point and two-point correlations have a long history, starting from the 80's [4] and continuing even to this day. It is impossible to review this area of research, instead we just mention a few key developments, mostly focusing on the XXZ and XYZ Heisenberg chains.

Most of the advances in the last three decades go back to the discoveries of the Izergin-Korepin [24,25] and Slavnov-determinants [26]. Exact results for form factors and correlation functions in the XXZ model were developed afterwards by the Lyon group [27–29], culminating in various multiple integral formulas for static and dynamical correlation functions. Eventually this led to the rigorous derivation of the asymptotic limit of correlation functions in various situations; for a review we refer the interested reader to the work [30]. Finite temperature correlation functions were worked out by the Wuppertal group, see for example [31–33]. It was also realized [34,35] that certain concrete multiple integral formulas for correlations of the XXX chain can be factorized. Afterwards this was developed into a full algebraic theory, which was presented in the literature in a series of works with contributions from many researchers [36–43].

Fewer results are available for models different from the XXZ spin chain. Correlation functions in the higher spin generalizations of the Heisenberg chain have been considered for example in [44–47]. Correlation functions in higher rank spin chains solvable by the nested Bethe Ansatz were considered for example in [48–53] and also in [54–56]. The results of [19] for the mean values of current operators in the $SU(3)$-symmetric model also belong to this list, together with certain factorized mean values found in [57] via lattice generalizations of the $T\bar{T}$-deformation known from quantum field theories.

The XYZ model appears exotic: there are fewer results available for this model, and this is due to the relative complexity of its exact solution. The diagonalization of the Hamiltonian goes back to Baxter [1], and it has been treated with various methods, focusing on various boundary conditions [58–61]. Bi-partite entanglement in the ground state in infinite volume was treated in [62]. Correlation functions in the so-called cyclic points were computed in [63],

and multiple integral formulas were computed in [64] and also in [65–67]. Finally, overlaps between Bethe states were considered recently in [68].

This paper consists of the following Sections: In Section 2 we formulate our main result for the XYZ chain, together with a quick review of the earlier result of the XXZ chains. In Section 3 we review the exact solution of the XYZ chain, and we prove our main result in Section 4. Section 5 contains our Conclusions and a few open problems. Finally, some details about the elliptic functions and the charge and current operators, and some numerical checks are presented in Appendices A-C.

## 2 Main result

In this work, we consider the spin-1/2 XYZ model in a finite size $L$, given by the Hamiltonian:

$$H = \sum_{j=1}^{L} h_{j,j+1} = \frac{1}{2} \sum_{j=1}^{L} \left[ J_x \sigma_j^x \sigma_{j+1}^x + J_y \sigma_j^y \sigma_{j+1}^y + J_z \sigma_j^z \sigma_{j+1}^z \right], \tag{1}$$

where $J_x$, $J_y$ and $J_z$ are real parameters of the model, while $\sigma_j^\alpha$ ($\alpha = x, y, z$) is the appropriate Pauli matrix acting on site $j$. We assume periodic boundary condition, so $\sigma_{L+1}^\alpha = \sigma_1^\alpha$. Moreover, we only consider the case of even $L$, for reasons that will become clear in Section 4. The case when $J_x$, $J_y$ and $J_z$ are all unequal is the XYZ model, while the cases $J_x = J_y = 1$, $J_z = \Delta$ and $J_x = J_y = J_z = 1$ correspond to the Heisenberg XXZ and XXX models, respectively. The Hamiltonian (1) is integrable for arbitrary values of the $J$'s, but integrability is broken with arbitrary magnetic fields if all $J$'s are different [69].

One of the key signs of integrability is that there exists a family of conserved, local operators $\hat{Q}_\alpha$, indexed by $\alpha$, which mutually commute with each other:

$$[\hat{Q}_\alpha, \hat{Q}_\beta] = 0, \tag{2}$$

and the Hamiltonian is a member of this family. However, while in the XXZ model the z-component of the total spin $S^z$ is also conserved, this is not the case in the XYZ model. Therefore the XYZ model lacks $U(1)$ symmetry, which makes it considerably harder to treat, as we shall see. Locality of the charge operators $\hat{Q}_\alpha$ means that they can be written as a sum of local charge densities, $\hat{q}_\alpha(j)$:

$$\hat{Q}_\alpha = \sum_{j=1}^{L} \hat{q}_\alpha(j), \tag{3}$$

where $\hat{q}_\alpha(j)$ acts non-trivially only on $\alpha$ sites, and is centered on site $j$. The construction of these charges by the means of the Quantum Inverse Scattering Method is presented in Section 3. An alternative method, using the boost operator is given in Appendix B, along with the explicit form of the first few charge operators. We should also note that explicit formulas for all conserved charges of the XYZ model were derived in [70] (for a related work see [71]).

Since the Hamiltonian is an element of the set of mutually commuting operators, the global charges $\hat{Q}_\alpha$ are conserved:

$$\frac{d}{dt} \hat{Q}_\alpha = i[H, \hat{Q}_\alpha] = 0. \tag{4}$$

However, in non-equilibrium situations, described by GHD, charges contained in a finite segment of the chain are of interest. The time evolution of these charge segments are given by

the continuity equation:

$$\frac{d}{dt}\sum_{j=j_1}^{j_2}\hat{q}_\alpha(j) = i\left[H,\sum_{j=j_1}^{j_2}\hat{q}_\alpha(j)\right] = \hat{J}_\alpha(j_1) - \hat{J}_\alpha(j_2+1). \tag{5}$$

This equation defines the physical current operator $\hat{J}_\alpha(j)$ corresponding to the charge $\hat{Q}_\alpha$. Similarly, one can define generalized current operators by considering the time evolution dictated by not the Hamiltonian $H$, but an other member of the family of commuting charges, $\hat{Q}_\beta$:

$$i\left[\hat{Q}_\beta,\sum_{j=j_1}^{j_2}\hat{q}_\alpha(j)\right] = \hat{J}_\alpha^\beta(j_1) - \hat{J}_\alpha^\beta(j_2+1). \tag{6}$$

Our goal is to compute the mean values $\langle n|\hat{J}_\alpha(j)|n\rangle$ and $\langle n|\hat{J}_\alpha^\beta(j)|n\rangle$, where $|n\rangle$ is an arbitrary eigenstate of the Hamiltonian (1). In models, such as the Heisenberg XXX and XXZ spin chains, where the exact eigenstates are found by the coordinate Bethe Ansatz method, compact formulae for the current mean values were already obtained by the authors and a collaborator [18–20]. First, we recall these results.

## 2.1 Current mean values in models with a simple Bethe Ansatz description

The results of [18–20] concern models with a global $U(1)$-symmetry. The primary examples are the XXZ Heisenberg spin chains and the Lieb-Liniger model, where the Bethe Ansatz takes a relatively simple form.

In these models, the (un-normalized) eigenstates can be constructed as

$$\Psi(x_1,\ldots,x_N) = \sum_{\sigma\in S_N}\left[\exp\left(i\sum_{j=1}^N p_{\sigma_j}x_j\right)\prod_{j<k}S(p_{\sigma_j},p_{\sigma_k})\right]. \tag{7}$$

Here $S(p_j,p_k)$ is the model-dependent two particle scattering amplitude. The wave function (7) is an eigenstate, given that the particle momenta $\{p_j\}_{j=1}^N$ satisfy the Bethe equations, coming from the periodic boundary condition:

$$e^{ip_jL}\prod_{k\neq j}S(p_j,p_k) = 1, \qquad j=1,\ldots,N. \tag{8}$$

It is often useful to introduce the rapidity parametrization $p_j = p(\lambda_j)$, in which the scattering amplitude is additive: $S(p_j,p_k) = S\big(p(\lambda_j),p(\lambda_k)\big) = S(\lambda_j-\lambda_k)$. The eigenstates of the system then described by the Bethe roots $\{\lambda_j\}_{j=1}^N$, which solve the Bethe equations. For the normalized $N$ particle Bethe states, we use the notation $|\lambda_1,\ldots,\lambda_N\rangle$. The total energy and momenta in these Bethe states are given additively:

$$E = \sum_{j=1}^N e(\lambda_j),$$
$$P = \sum_{j=1}^N p(\lambda_j) \mod 2\pi, \tag{9}$$

where $e(\lambda)$ is the model dependent one-particle energy. Similarly, the eigenvalues of the conserved charge operators are also calculated additively:

$$\hat{Q}_\alpha|\lambda_1,\ldots,\lambda_N\rangle = \left(\sum_{j=1}^N q_\alpha(\lambda_j)\right)|\lambda_1,\ldots,\lambda_N\rangle, \tag{10}$$

where $q_\alpha(\lambda)$ is the one-particle eigenvalue. In the calculation of the current mean values, the so-called Gaudin matrix plays an important role. To define this quantity, first we write the Bethe equations in logarithmic form:

$$p(\lambda_j)L + \sum_{k \neq j} \delta(\lambda_j - \lambda_k) = 2\pi I_j, \qquad j = 1, \ldots, N, \tag{11}$$

where $\delta(\lambda) = -i \log S(\lambda)$, and $I_j \in \mathbb{Z}$ are the momentum quantum numbers. The Gaudin matrix is then defined as:

$$G_{jk} = \frac{\partial}{\partial \lambda_k}\big(2\pi I_j\big), \tag{12}$$

or written out explicitly:

$$G_{jk} = \delta_{jk}\left[p'(\lambda_j)L + \sum_{l=1}^{N} \varphi(\lambda_j - \lambda_l)\right] - \varphi(\lambda_j - \lambda_k), \tag{13}$$

where $\varphi(\lambda) = \delta'(\lambda)$. The mean values of the current and generalized current operators can be calculated as:

$$\langle \lambda_1, \ldots, \lambda_N | \hat{J}_\alpha(j) | \lambda_1, \ldots, \lambda_N \rangle = \mathbf{e}' \cdot G^{-1} \cdot \mathbf{q}_\alpha, \tag{14}$$

and

$$\langle \lambda_1, \ldots, \lambda_N | \hat{J}_\alpha^\beta(j) | \lambda_1, \ldots, \lambda_N \rangle = \mathbf{q}'_\beta \cdot G^{-1} \cdot \mathbf{q}_\alpha. \tag{15}$$

Here $\mathbf{e}'$, $\mathbf{q}_\alpha$ and $\mathbf{q}'_\beta$ are $N$-component vectors with elements

$$(\mathbf{e}')_j = \frac{de(\lambda_j)}{d\lambda}, \qquad (\mathbf{q}_\alpha)_j = q_\alpha(\lambda_j), \qquad (\mathbf{q}'_\beta)_j = \frac{dq_\beta(\lambda_j)}{d\lambda}, \tag{16}$$

and $G^{-1}$ is the inverse of the Gaudin matrix. Formulae (14) and (15) were first obtained in [18] by the use of a form factor expansion. Later, the results were extended to models solvable by the nested Bethe Ansatz, using a connection to long range spin chains [19]. Finally, in [20] the algebraic construction of current operators was given and (14) and (15) were re-derived.

In [18] a semi-classical interpretation of the results was also given, based on the ideas of [17]. In this semi-classical picture, the system is considered as a ring with circumference $L$, on which $N$ particles move around. Particles travel along with constant velocity as long as they don't interact with each other. A particle with rapidity $\lambda$, represented by a wave packet has a bare velocity

$$v(\lambda) = \frac{de}{dp} = \frac{e'(\lambda)}{p'(\lambda)}. \tag{17}$$

Since these speeds are generally different for different rapidities, the particles meet each other along the way. Their scattering can be taken into account by using the exact quantum mechanical solution of the two-body problem: for the scattering of particle $j$ on $k$ the wave function has to be multiplied by the phase $S(\lambda_j - \lambda_k)$. As a result of this rapidity dependent phase, the wave packets suffer a spatial displacement (or equivalently a time delay), given by the derivative of the scattering phase $\delta$, with respect to the momentum. After the scattering, the particles continue to travel with their bare velocities. Since the particles move along a closed ring, they scatter on each other over and over again. The time delays picked up by each scattering event eventually result in the emergence of an effective velocity $v_{\text{eff}}(\lambda)$. Since a particle with rapidity $\lambda$ carries the charge eigenvalue $q_\alpha(\lambda)$, the current mean value in this semi-classical picture is simple given by

$$J_{\alpha,cl} = \frac{1}{L} \sum_{j=1}^{N} v_{\text{eff}}(\lambda_j) q_\alpha(\lambda_j). \tag{18}$$

A simple self-consistent calculation shows, that the effective velocities can be calculated as

$$\mathbf{v}_{\text{eff}} = L G^{-1} \mathbf{e}'.\tag{19}$$

Substituting this $v_{\text{eff}}(\lambda)$ into (18) reproduces the previous result (14). Thus, in these Bethe Ansatz solvable models a complete classical/quantum correspondence is present.

## 2.2 Main results for the XYZ spin chain

In models lacking $U(1)$ symmetry, the standard coordinate Bethe Ansatz breaks down and the simple semi-classical picture cannot be applied. Therefore it is an interesting question what happens with the current mean values.

The main result of the present work is that an expression completely analogous to (14) and (15) holds even in the case of the XYZ model:

$$\langle \lambda_1, \ldots, \lambda_n | \hat{J}^{\beta}_{\alpha}(j) | \lambda_1, \ldots, \lambda_n \rangle = \mathbf{q}'_{\beta} \cdot G^{-1} \cdot \mathbf{q}_{\alpha}.\tag{20}$$

Here $|\lambda_1, \ldots, \lambda_n\rangle$ is an eigenstate of the XYZ model, which can be constructed using a generalization of the Algebraic Bethe Ansatz, and is described by a fixed number of rapidities (precisely $n = L/2$ for all eigenstates.) These rapidities again satisfy a system of Bethe equations, and the Gaudin matrix is defined as the logarithmic derivative of these Bethe equations. The vectors $\mathbf{q}'_{\beta}$ and $\mathbf{q}_{\alpha}$ contain the charge eigenvalues, just like in (16). The quantities in (20) are introduced in more detail in Sections 3 and 4.

The existence of the formula (20) shows, that even though the XYZ model lacks $U(1)$ symmetry and the semi-classical picture presented above is not applicable, the current mean values still take a simple form, and are well described by the rapidity variables. The proof of (20) is presented in Section 4.

# 3 Integrability of the XYZ model

In this section, we review how the XYZ model fits into the standard framework of the Quantum Inverse Scattering Method, and how its eigenstates can be constructed, using a generalization of the well-known Algebraic Bethe Ansatz.

## 3.1 Transfer matrix

In the QISM framework, the $R$-matrix $R(\lambda) \in \text{End}(\mathbb{C}^2 \otimes \mathbb{C}^2)$ associated to the eight vertex model is given by:

$$R(\lambda) = \frac{1}{h(\lambda + \eta)} \begin{pmatrix} a(\lambda) & 0 & 0 & d(\lambda) \\ 0 & b(\lambda) & c(\lambda) & 0 \\ 0 & c(\lambda) & b(\lambda) & 0 \\ d(\lambda) & 0 & 0 & a(\lambda) \end{pmatrix},\tag{21}$$

where the functions $a(\lambda), b(\lambda), c(\lambda), d(\lambda)$ and $h(\lambda)$ are defined as:

$$\begin{aligned}
a(\lambda) &= \vartheta_4(\eta|\tau)\vartheta_1(\lambda + \eta|\tau)\vartheta_4(\lambda|\tau), \\
b(\lambda) &= \vartheta_4(\eta|\tau)\vartheta_4(\lambda + \eta|\tau)\vartheta_1(\lambda|\tau), \\
c(\lambda) &= \vartheta_1(\eta|\tau)\vartheta_4(\lambda + \eta|\tau)\vartheta_4(\lambda|\tau), \\
d(\lambda) &= \vartheta_1(\eta|\tau)\vartheta_1(\lambda + \eta|\tau)\vartheta_1(\lambda|\tau), \\
h(\lambda) &= \vartheta_4(0|\tau)\vartheta_4(\lambda|\tau)\vartheta_1(\lambda|\tau).
\end{aligned}\tag{22}$$

Here $\eta \in \mathbb{C}$ is a parameter of the model, while $\vartheta_i(\lambda|\tau)$ ($i \in \{1, 2, 3, 4\}$) denote the elliptic theta functions with parameter $\tau$ and argument $\lambda$. The definition of these functions and a list of their useful properties are presented in Appendix A. For brevity, in the following we omit the dependence on the parameter $\tau$, and simply write $\vartheta_i(\lambda) = \vartheta_i(\lambda|\tau)$. The $R$-matrix (21) satisfies the usual Yang-Baxter equation, acting on the triple product space $\mathbb{C}^2 \otimes \mathbb{C}^2 \otimes \mathbb{C}^2$:

$$R_{12}(\lambda_{12})R_{13}(\lambda_{13})R_{23}(\lambda_{23}) = R_{23}(\lambda_{23})R_{13}(\lambda_{13})R_{12}(\lambda_{12}), \tag{23}$$

as well as the so-called regularity, unitary and crossing relations:

$$
\begin{aligned}
R_{12}(0) &= \mathcal{P}_{12}, \\
R_{21}(-\lambda)R_{12}(\lambda) &= \mathbb{1}, \\
R_{12}(\lambda)\sigma_1^y[R_{12}(\lambda - \eta)]^{t_1}\sigma_1^y &= \frac{h(\lambda - \eta)}{h(\lambda)}\mathbb{1}.
\end{aligned}
\tag{24}
$$

Here $\mathcal{P}$ and $\mathbb{1}$ are the permutation and identity operators respectively, and we also used the shorthand notation $\lambda_{ij} = \lambda_i - \lambda_j$, while $[\dots]^{t_1}$ denotes partial transpose with respect to the first space.

Using the $R$-matrix, the monodromy matrix $T_a(\lambda)$ can be constructed:

$$T_a(\lambda) = R_{aL}(\lambda)R_{aL-1}(\lambda)\dots R_{a1}(\lambda). \tag{25}$$

This operator acts on the product space $V_a \otimes \mathcal{H}$, where $\mathcal{H} = \bigotimes_{j=1}^{L} \mathbb{C}_j^2$ is the physical space, while $V_a$ is an auxiliary space, which in our case is also isomorphic to $\mathbb{C}^2$. As a consequence of the Yang-Baxter equation, the monodromy matrix satisfies the so-called $RTT$-relation:

$$R_{ab}(u - v)T_a(u)T_b(v) = T_b(v)T_a(u)R_{ab}(u - v), \tag{26}$$

where now $a$ and $b$ denote two different auxiliary spaces. By taking the partial trace of the monodromy matrix over the auxiliary space, the transfer matrix $t(\lambda)$ – acting on the physical space $\mathcal{H}$ – can be defined:

$$t(\lambda) = \text{Tr}_a T_a(\lambda). \tag{27}$$

From the $RTT$-relation (26), it follows, that transfer matrices at different values of the rapidity parameter commute with each other:

$$[t(u), t(v)] = 0. \tag{28}$$

This property ensures that a commuting set of local charges is obtained by expanding the logarithm of the transfer matrix around zero:

$$\hat{Q}_\alpha = (i)^{\alpha-2}\frac{d^{\alpha-1}}{d\lambda^{\alpha-1}}\left(\log t(\lambda)\right)\Big|_{\lambda=0}. \tag{29}$$

(Here the factor of $(i)^{\alpha-2}$ is included to ensure the hermiticity of the charges.) Specifically, the Hamiltonian of the XYZ model (1) is given by

$$\frac{d}{d\lambda}\left(\log t(\lambda)\right)\Big|_{\lambda=0} = H - \frac{1}{2}LJ_0, \tag{30}$$

with the coefficients in the Hamiltonian defined as

$$
\begin{aligned}
J_x &= \frac{\vartheta_1'(0)}{\vartheta_4(0)}\left(\frac{\vartheta_4(\eta)}{\vartheta_1(\eta)} + \frac{\vartheta_1(\eta)}{\vartheta_4(\eta)}\right), &\qquad J_z &= \frac{\vartheta_1'(\eta)}{\vartheta_1(\eta)} - \frac{\vartheta_4'(\eta)}{\vartheta_4(\eta)}, \\
J_y &= \frac{\vartheta_1'(0)}{\vartheta_4(0)}\left(\frac{\vartheta_4(\eta)}{\vartheta_1(\eta)} - \frac{\vartheta_1(\eta)}{\vartheta_4(\eta)}\right), &\qquad J_0 &= \frac{\vartheta_1'(\eta)}{\vartheta_1(\eta)} + \frac{\vartheta_4'(\eta)}{\vartheta_4(\eta)}.
\end{aligned}
\tag{31}
$$

## 3.2  Generalized algebraic Bethe Ansatz

As opposed to the XXX or XXZ spin chains, in the case of the XYZ model, the family of commuting conserved charges does not contain the $S_z$ operator, therefore this model lacks $U(1)$ symmetry. As a result, the usual methods of Coordinate/Algebraic Bethe Ansatz break down. However, in [58] a generalization of the well-known Algebraic Bethe Ansatz was introduced and successfully applied to the eight vertex model, making it possible to construct the transfer matrix eigenstates. Here, we briefly review this method.

In the usual Algebraic Bethe Ansatz framework, one considers the monodromy matrix as a $2 \times 2$ matrix in the auxiliary space, with elements acting on the physical space:

$$T_a(\lambda) = \begin{pmatrix} A(\lambda) & B(\lambda) \\ C(\lambda) & D(\lambda) \end{pmatrix}, \tag{32}$$

where $A(\lambda), B(\lambda), C(\lambda), D(\lambda) \in \mathrm{End}(\mathcal{H})$. In order to obtain the eigenstates, first a global reference state $|0\rangle$ is chosen, which is an eigenstate of the diagonal elements of the monodromy matrix and is annihilated by its bottom left element:

$$A(\lambda)|0\rangle = \Lambda(\lambda)|0\rangle, \qquad D(\lambda)|0\rangle = \tilde{\Lambda}(\lambda)|0\rangle, \qquad C(\lambda)|0\rangle = 0. \tag{33}$$

Then the Bethe states are constructed by applying the top right element of the monodromy matrix on this reference state:

$$|\lambda_1, \ldots, \lambda_n\rangle = B(\lambda_1) \ldots B(\lambda_n)|0\rangle. \tag{34}$$

Finally, by using the commutation relations between monodromy matrix elements, contained in the $RTT$-relation (26), it is shown, that the state (34) is indeed an eigenstate of the transfer matrix $t(\lambda) = A(\lambda) + D(\lambda)$, if the set of rapidities $\{\lambda_1, \ldots, \lambda_n\}$ satisfies a system of non-linear equations, known as the Bethe equations.

The problem with the method explained above, is that in the case of the XYZ model, there is no such global reference state $|0\rangle$ which is annihilated by $C(\lambda)$, for all values of $\lambda$. To overcome this problem, in [58] a gauge transformation was applied on the $R$-matrix, which makes it possible to introduce a family of reference states.

First, we consider the local $R$-matrix as a $2 \times 2$ matrix in the auxiliary space, with elements acting on the local physical space $\mathbb{C}^2$:

$$R_{aj}(\lambda) = \begin{pmatrix} \alpha_j(\lambda) & \beta_j(\lambda) \\ \gamma_j(\lambda) & \delta_j(\lambda) \end{pmatrix}, \tag{35}$$

with $\alpha_j(\lambda), \beta_j(\lambda), \gamma_j(\lambda), \delta_j(\lambda) \in \mathrm{End}(\mathbb{C}_j^2)$. Now let's introduce a local gauge transformation in the following way:

$$R_{aj}^l(\lambda) = M_{j+l}^{-1}(\lambda) R_{aj}(\lambda) M_{j+l-1}(\lambda) = \begin{pmatrix} \alpha_j^l(\lambda) & \beta_j^l(\lambda) \\ \gamma_j^l(\lambda) & \delta_j^l(\lambda) \end{pmatrix}, \tag{36}$$

where $M_k$ is a $2 \times 2$ invertible, numerical matrix and $l$ is an integer. Obviously, the monodromy matrix built from the transformed $R$-matrix differs from the original one by a simple linear transformation:

$$T_a^l(\lambda) = R_{aL}^l(\lambda) \ldots R_{a1}^l(\lambda) = M_{L+l}^{-1}(\lambda) T_a(\lambda) M_l(\lambda) = \begin{pmatrix} A_l(\lambda) & B_l(\lambda) \\ C_l(\lambda) & D_l(\lambda) \end{pmatrix}. \tag{37}$$

As it turns out, $M_k(\lambda)$ can be chosen in such a way, that there is a local reference state at each site, which is annihilated by the bottom left element of the corresponding local matrix $R_{aj}^l(\lambda)$, for all values of $\lambda$. To achieve this, we define $M_k(\lambda)$ as follows:

$$M_k(\lambda; s, t) = \begin{pmatrix} \vartheta_1(s + k\eta - \lambda) & \frac{1}{g(\tau_k)} \vartheta_1(t + k\eta + \lambda) \\ \vartheta_4(s + k\eta - \lambda) & \frac{1}{g(\tau_k)} \vartheta_4(t + k\eta + \lambda) \end{pmatrix}, \tag{38}$$

where $g(u) = \vartheta_1(u)\vartheta_4(u)$, $\tau_k = \frac{s+t}{2} + k\eta - \pi/2$ and $s$ and $t$ are two arbitrary, but fixed complex parameters, which we will not denote in the following. By using the addition theorems for the theta functions listed in Appendix A, it is easy to check that the local state $\omega_j^l \in \mathbb{C}_j^2$, defined as

$$\omega_j^l = \vartheta_1(s + (j+l)\eta)e_j^+ + \vartheta_4(s + (j+l)\eta)e_j^-, \tag{39}$$

is annihilated by $\gamma_j^l(\lambda)$ for all $\lambda$:

$$\gamma_j^l(\lambda)\omega_j^l = 0. \tag{40}$$

Here $e_j^{+/-}$ are the standard basis elements at site $j$. Similarly, the actions of $\alpha_j^l(\lambda)$ and $\delta_j^l(\lambda)$ on $\omega_j^l$ are also easy to compute:

$$\begin{aligned} \alpha_j^l(\lambda)\omega_j^l &= \omega_j^{l-1}, \\ \delta_j^l(\lambda)\omega_j^l &= \frac{h(\lambda)}{h(\lambda+\eta)}\omega_j^{l+1}. \end{aligned} \tag{41}$$

From the local formulae (40) and (41), it follows that the actions of the transformed monodromy matrix elements on the global state $\Omega_l \in \mathcal{H}$ defined as

$$\Omega_l = \omega_1^l \otimes \omega_2^l \otimes \cdots \otimes \omega_L^l, \tag{42}$$

are given by

$$\begin{aligned} A_l(\lambda)\Omega_l &= \Omega_{l-1}, \\ D_l(\lambda)\Omega_l &= \left(\frac{h(\lambda)}{h(\lambda+\eta)}\right)^L \Omega_{l+1}, \\ C_l(\lambda)\Omega_l &= 0. \end{aligned} \tag{43}$$

This means, that the vectors $\{\Omega_l\}_{l\in\mathbb{Z}}$ form a family of reference states, on which the actions of the transformed monodromy matrix elements are known. In order to construct the eigenstates of the model, it is useful to consider more general transformations of the monodromy matrix than the one defined in (37), by not restricting the indexes of the transformation matrices:

$$T_a^{k,l}(\lambda) = M_k^{-1}(\lambda)T_a(\lambda)M_l(\lambda) = \begin{pmatrix} A_{k,l}(\lambda) & B_{k,l}(\lambda) \\ C_{k,l}(\lambda) & D_{k,l}(\lambda) \end{pmatrix}. \tag{44}$$

As it turns out, the commutation relations between $A_{k,l}(\lambda), B_{k,l}(\lambda), C_{k,l}(\lambda)$ and $D_{k,l}(\lambda)$ take a simple form. These relations can be derived from the $RTT$-relation (26), using the known form of the transformation matrix (38) and the properties of the elliptic functions. The actual computation is rather long and technical, therefore we only present here the formulae that are used in further calculations. For more details, we refer to [58]. The commutation relations necessary for us are given by:

$$\begin{aligned} B_{k,l+1}(\lambda)B_{k+1,l}(\mu) &= B_{k,l+1}(\mu)B_{k+1,l}(\lambda), \\ A_{k,l}(\lambda)B_{k+1,l-1}(\mu) &= \alpha(\lambda,\mu)B_{k,l-2}(\mu)A_{k+1,l-1}(\lambda) - \beta_{l-1}(\lambda,\mu)B_{k,l-2}(\lambda)A_{k+1,l-1}(\mu), \\ D_{k,l}(\lambda)B_{k+1,l-1}(\mu) &= \alpha(\mu,\lambda)B_{k+2,l}(\mu)D_{k+1,l-1}(\lambda) + \beta_{k+1}(\lambda,\mu)B_{k+2,l}(\lambda)D_{k+1,l-1}(\mu), \end{aligned} \tag{45}$$

where the functions $\alpha(\lambda,\mu)$ and $\beta(\lambda,\mu)$ are defined as:

$$\alpha(\lambda,\mu) = \frac{h(\lambda-\mu-\eta)}{h(\lambda-\mu)}, \qquad \beta_k(\lambda,\mu) = \frac{h(\eta)h(\tau_k+\mu-\lambda)}{h(\mu-\lambda)h(\tau_k)}. \tag{46}$$

At this point, we are ready to construct the transfer matrix eigenstates. First, let's examine the vector

$$\Psi_l(\lambda_1,\ldots,\lambda_n) = B_{l+1,l-1}(\lambda_1)\ldots B_{l+n,l-n}(\lambda_n)\Omega_{l-n}, \tag{47}$$

where the rapidities $\{\lambda_k\}_{k=1}^n$ are not yet specified, and $n = L/2$. From (44) it is obvious, that $t(\lambda) = A(\lambda) + D(\lambda) = A_{l,l}(\lambda) + D_{l,l}(\lambda)$ for any $l$. By using the commutation relations, the actions of $A_{l,l}(\lambda)$ and $D_{l,l}(\lambda)$ on the state (47) are easily obtained:

$$A_{l,l}(\lambda)\Psi_l(\lambda_1,\ldots,\lambda_n) = \Lambda(\lambda|\lambda_1,\ldots,\lambda_n)\Psi_{l-1}(\lambda_1,\ldots,\lambda_n)$$
$$+ \sum_{j=1}^n \Lambda_j^l(\lambda|\lambda_1,\ldots,\lambda_n)\Psi_{l-1}(\lambda_1,\ldots,\lambda_{j-1},\lambda,\lambda_{j+1},\ldots,\lambda_n),$$
$$D_{l,l}(\lambda)\Psi_l(\lambda_1,\ldots,\lambda_n) = \tilde{\Lambda}(\lambda|\lambda_1,\ldots,\lambda_n)\Psi_{l+1}(\lambda_1,\ldots,\lambda_n) \tag{48}$$
$$+ \sum_{j=1}^n \tilde{\Lambda}_j^l(\lambda|\lambda_1,\ldots,\lambda_n)\Psi_{l+1}(\lambda_1,\ldots,\lambda_{j-1},\lambda,\lambda_{j+1},\ldots,\lambda_n),$$

with

$$\Lambda(\lambda|\lambda_1,\ldots,\lambda_n) = \prod_{k=1}^n \alpha(\lambda,\lambda_k),$$
$$\Lambda_j^l(\lambda|\lambda_1,\ldots,\lambda_n) = -\beta_{l-1}(\lambda,\lambda_j)\prod_{\substack{k=1\\k\neq j}}^n \alpha(\lambda_j,\lambda_k),$$
$$\tilde{\Lambda}(\lambda|\lambda_1,\ldots,\lambda_n) = \left(\frac{h(\lambda)}{h(\lambda+\eta)}\right)^L \prod_{k=1}^n \alpha(\lambda_k,\lambda), \tag{49}$$
$$\tilde{\Lambda}_j^l(\lambda|\lambda_1,\ldots,\lambda_n) = \beta_{l+1}(\lambda,\lambda_j)\left(\frac{h(\lambda_j)}{h(\lambda_j+\eta)}\right)^L \prod_{\substack{k=1\\k\neq j}}^n \alpha(\lambda_k,\lambda_j).$$

The condition $n = L/2$ is necessary, because after $A_{l,l}(\lambda)$ is commuted through all the $B(\lambda_k)$ operators in (47), finally the expression $A_{l+n,l-n}(\lambda)\Omega_{l-n}$ has to be evaluated. This is only possible by using (43), if $n = L/2$. The restriction on $n$ also means, that the method presented here only works for chains of even length.

As a last step, we multiply (47) with the factor $\exp(2\pi i l\theta)$ $(0 \le \theta \le 1)$ and sum it over all the integers:

$$\Psi_\theta(\lambda_1,\ldots,\lambda_n) = \sum_{l=-\infty}^\infty e^{2\pi i l\theta}\Psi_l(\lambda_1,\ldots,\lambda_n). \tag{50}$$

From (48) it follows that the state (50) is an eigenstate of the eight vertex transfer matrix, with eigenvalue

$$\Lambda_f(\lambda) = e^{2\pi i\theta}\Lambda(\lambda|\lambda_1,\ldots,\lambda_n) + e^{-2\pi i\theta}\tilde{\Lambda}(\lambda|\lambda_1,\ldots,\lambda_n), \tag{51}$$

given that the rapidities satisfy the Bethe equations:

$$\left(\frac{h(\lambda_j)}{h(\lambda_j+\eta)}\right)^L = e^{4\pi i\theta}\prod_{\substack{k=1\\k\neq j}}^n \frac{\alpha(\lambda_j,\lambda_k)}{\alpha(\lambda_k,\lambda_j)}. \tag{52}$$

Since the Hamiltonian of the XYZ model is connected to the transfer matrix through (30), it is obvious, that the states (50) are also eigenstates of the Hamiltonian (1). The energy eigenvalues are given by

$$E = \frac{d}{d\lambda}\Big(\log t(\lambda)\Big)\Big|_{\lambda=0} + \frac{LJ_0}{2}. \tag{53}$$

Equivalently, we can write the energies (53) as sums over the Bethe roots:

$$E = \sum_{j=1}^{n} e(\lambda_j) + \frac{LJ_0}{2}, \quad \text{with} \quad e(\lambda) = \frac{h'(\lambda)}{h(\lambda)} - \frac{h'(\lambda + \eta)}{h(\lambda + \eta)}. \tag{54}$$

Furthermore, from (29) it follows, that the eigenvalues of the higher charges also can be written as sums over the rapidities:

$$\hat{Q}_\alpha |\lambda_1, \ldots, \lambda_n\rangle = \left( \sum_{j=1}^{n} q_\alpha(\lambda_j) \right) |\lambda_1, \ldots, \lambda_n\rangle, \quad \text{with} \quad q_\alpha(\lambda) = (-i)^{\alpha-2} \frac{d^{\alpha-2}}{d\lambda^{\alpha-2}} e(\lambda). \tag{55}$$

The convergence of the sum in (50) requires a careful analysis. The results of [1] and [58] show, that (50) is summable to zero for all $\theta$ expect finitely many values of $\theta_j$, and that 0 is among these $\theta_j$. Nevertheless, our derivation for the current mean values does not depend on the value of $\theta$. The situation simplifies if we impose the following restriction on $\eta$:

$$K\eta = m_1 2\pi + m_2 \pi \tau, \tag{56}$$

with $K$, $m_1$ and $m_2$ being integers. Because of the quasi-periodicity of the elliptic theta functions, $M_k(\lambda)$, $T_a^{k,l}(\lambda)$ and $\Psi_l(\lambda_1, \ldots, \lambda_n)$ are quasi-periodic functions of $k$ and $l$ in this case. With an appropriate choice of common normalizing factor, it can be arranged that they become periodic in $k$ and $l$ with period $K$. As a result, the sum in (50) becomes finite, with the values of $\theta$ given as $\theta = k/K$ for $k = 0, 1, \ldots, K-1$. For more details on this special choice of $\eta$ see [58]. However, in the following we consider the general case, with no restriction on $\eta$.

## 4 Proof of current mean value formula

In this section, we present the proof of our main result for the current mean values. The proof is based on the algebraic construction of current operators, that was worked out in [20]. In that paper, it was shown in a model independent way, how the current operators of integrable spin chains can be embedded into the usual framework of Yang-Baxter integrability. Based on this algebraic construction, it was also shown that the current mean values are related to the eigenvalues of a transfer matrix, defined on an enlarged chain. Here, we evaluate this transfer matrix eigenvalue, by using the method explained in Section 3 and prove (20).

### 4.1 Algebraic construction of the current operators

The construction of the conserved charge operators of integrable systems by the means of the Quantum Inverse Scattering Method has been known for a long time. However, the current operators were only embedded into this framework recently by one of the authors [20]. Our proof of (20) is built upon this algebraic construction, therefore here we briefly summarize the results of [20].
Here, we consider a generic integrable system, corresponding to an $R$-matrix $R(\lambda)$, that satisfies the Yang-Baxter equation (23) and the regularity and unitarity conditions (24). The monodromy and transfer matrices of the model are defined as in (25) and (27), respectively. Then a generating function for the global charges is defined as:

$$\hat{Q}(\lambda) = (-i) t^{-1}(\lambda) \frac{d}{d\lambda} t(\lambda). \tag{57}$$

This definition agrees with (29) (up to factors of $i$), the canonical charges are given as the coefficients of the Taylor expansion of $\hat{Q}(\lambda)$, around $\lambda = 0$. A charge density for this generating

function can be found: $\hat{Q}(\lambda) = \sum_{j=1}^{L} \hat{q}(\lambda, j)$ with

$$\hat{q}(\lambda, j) = (-i)t^{-1}(\lambda)Tr_a\big[T_a^{[L,j+1]}(\lambda)\partial_\lambda R_{a,j}(\lambda)T_a^{[j-1,1]}(\lambda)\big]. \tag{58}$$

Here $T_a^{[j_2,j_1]}(\lambda)$ is a partial monodromy matrix defined on the segment $[j_1,\ldots,j_2]$:

$$T_a^{[j_2,j_1]}(\lambda) = R_{a,j_2}(\lambda)\ldots R_{a,j_1}(\lambda). \tag{59}$$

Similarly to the charge operators, we can define a generating operator for the currents as well:

$$\hat{J}(\lambda, \mu, j) = \sum_{\alpha=2}^{\infty}\sum_{\beta=2}^{\infty}\frac{\lambda^{\alpha-2}}{(\alpha-2)!}\frac{\mu^{\beta-2}}{(\beta-2)!}\hat{J}_\alpha^\beta(j), \tag{60}$$

where $\hat{J}_\alpha^\beta(j)$ are the generalized currents, defined in (6). The generating functions $\hat{Q}(\lambda)$ and $\hat{J}(\lambda, \mu, j)$ satisfy the generalized continuity equation:

$$i\big[\hat{Q}(\lambda), \hat{q}(\mu, j)\big] = \hat{J}(\lambda, \mu, j) - \hat{J}(\lambda, \mu, j+1). \tag{61}$$

Using this continuity equation and the Yang-Baxter equation (23), it can be shown, that the current generating function is given by the following expression:

$$\hat{J}(\lambda, \mu, j) = -t(\mu)\partial_\mu\hat{\Omega}(\lambda, \mu, j-1)t^{-1}(\mu), \tag{62}$$

where $\hat{\Omega}(\lambda, \mu, j)$ is a „double-row" operator defined as

$$\hat{\Omega}(\lambda, \mu, j) = t^{-1}(\mu)t^{-1}(\lambda)Tr_{ab}[T_a^{[L,j+1]}(\lambda)T_b^{[L,j+1]}(\mu)\Theta_{a,b}(\lambda, \mu)T_a^{[j,1]}(\lambda)T_b^{[j,1]}(\mu)]. \tag{63}$$

Here $a$ and $b$ are two different auxiliary spaces, and $\Theta_{a,b}(\lambda, \mu)$ is an operator acting on these auxiliary spaces:

$$\Theta_{a,b}(\lambda, \mu) = (-i)R_{b,a}(\mu, \lambda)\partial_\lambda R_{a,b}(\lambda, \mu). \tag{64}$$

From (62) it simply follows that the mean values of the current generator in the eigenstates of the model are given by:

$$\langle\Psi|\hat{J}(\lambda, \mu, j)|\Psi\rangle = -\partial_\mu\langle\Psi|\hat{\Omega}(\lambda, \mu, j-1)|\Psi\rangle. \tag{65}$$

Expressions (62) and (65) show, how the current operators and their mean values can be constructed in the QISM framework. Moreover, using a trick first developed in [72], the mean values of the operator $\hat{\Omega}(\lambda, \mu, j)$ can be related to the eigenvalues of a transfer matrix of an enlarged spin chain. Let's consider the following monodromy matrix:

$$T_a^+(\lambda) = R_{a,L+2}(\lambda, \mu+\epsilon)R_{L+1,a}^{t_{L+1}}(\mu, \lambda)T_a(\lambda), \tag{66}$$

where $T_a(\lambda)$ is the original monodromy matrix of a chain of length $L$, and $[\ldots]^{t_{L+1}}$ denotes partial transposition with respect to the physical space at site $L+1$. Since the $R$-matrices introduced on the two extra sites also satisfy the Yang-Baxter relation, it implies that the transfer matrix defined as

$$t^+(\lambda) = Tr_a[T_a^+(\lambda)], \tag{67}$$

also forms a one parameter family of commuting operators. At $\epsilon = 0$ the two extra sites decouple from the rest of the chain, and the eigenstates of the enlarged transfer matrix take the following form:

$$t^+(\lambda)\big(|\delta\rangle \otimes |\Psi\rangle\big) = \Lambda_f(\lambda)\big(|\delta\rangle \otimes |\Psi\rangle\big), \tag{68}$$

where the „delta-state" $|\delta\rangle$ is given by the components $\delta_{i,j}$ in the computational basis, while $|\Psi\rangle$ is an eigenstate and $\Lambda_f(\lambda)$ is the corresponding eigenvalue of the original transfer matrix $t(\lambda)$. Switching on a non-zero $\epsilon$ affects the eigenstates and eigenvalues. Let $\left|\Psi^+\right\rangle$ be the eigenstate of $t^+(\lambda)$ that in the $\epsilon \to 0$ limit becomes $|\delta\rangle \otimes |\Psi\rangle$, and $\Lambda^+(\lambda|\mu,\epsilon)$ the eigenvalue corresponding to $\left|\Psi^+\right\rangle$. Then a first order perturbation theory computation gives that

$$\langle\Psi|\hat{\Omega}(\lambda,\mu,j)|\Psi\rangle = i\frac{d}{d\epsilon}\log\Lambda^+(\mu|\lambda,j)\bigg|_{\epsilon=0}. \tag{69}$$

Accordingly, the mean values of the current generating function are then given by:

$$\langle\Psi|\hat{J}(\lambda,\mu,j)|\Psi\rangle = -i\partial_\mu\left(\frac{d}{d\epsilon}\log\Lambda^+(\mu|\lambda,\epsilon)\bigg|_{\epsilon=0}\right). \tag{70}$$

Expression (70) shows that the current mean values are closely related to transfer matrix eigenvalues, which can be calculated by standard methods of integrability. For the case of the XXZ model, (70) was evaluated in [20]. In the next subsection, we extend this calculation to the case of the XYZ model.

## 4.2 Mean value formula

As it was explained in the previous subsection, the current mean values are closely related to an auxiliary eigenvalue problem of an enlarged transfer matrix. This eigenvalue problem can be treated by standard methods of integrability, which explains why simple formulas for the current mean values exist. In the remainder of this section, we solve this auxiliary eigenvalue problem and evaluate the necessary differentiation present in (70) to prove our main result (20). Our method closely follows the calculation presented in [20] for the case of the XXZ spin chain. The main difference is that instead of the usual Algebraic Bethe Ansatz technique, here we use the generalized method, explained in Section 3.

First, we introduce local gauge transformations for the extra two sites exactly the same way as before in (36), with the transformation matrix still given by (38). Then the enlarged, gauged transformed monodromy matrix is given by

$$T_a^{+,l}(\lambda) = M_{L+2+l}^{-1}(\lambda)T_a^+(\lambda)M_l(\lambda) = \begin{pmatrix} A_l^+(\lambda) & B_l^+(\lambda) \\ C_l^+(\lambda) & D_l^+(\lambda) \end{pmatrix}. \tag{71}$$

As a next step, local reference states for the extra two sites need to be defined. Based on the crossing relation (24) and the addition theorems for the theta functions, it is easily checked that the local state at site $L + 1$, defined as

$$\omega_{L+1}^l(\mu) = \vartheta_4(s + (L+l)\eta - \mu)e_{L+1}^+ - \vartheta_1(s + (L+l)\eta - \mu)e_{L+1}^-, \tag{72}$$

satisfies the following relations:

$$\begin{aligned} \alpha_{L+1}^l(\lambda)\omega_{L+1}^l(\mu) &= \frac{h(\lambda-\mu)}{h(\lambda-\mu-\eta)}\omega_{L+1}^{l-1}(\mu), \\ \delta_{L+1}^l(\lambda)\omega_{L+1}^l(\mu) &= \omega_{L+1}^{l+1}(\mu), \\ \gamma_{L+1}^l(\lambda)\omega_{L+1}^l(\mu) &= 0. \end{aligned} \tag{73}$$

Similarly, for site $L + 2$ the reference state can be chosen as

$$\omega_{L+2}^l(\mu,\epsilon) = \vartheta_1(s + (L+2+l)\eta - \mu - \epsilon)e_{L+2}^+ + \vartheta_4(s + (L+2+l)\eta - \mu - \epsilon)e_{L+2}^-. \tag{74}$$

Then the action of the local operators are given by:

$$
\begin{aligned}
\alpha_{L+2}^l(\lambda)\omega_{L+2}^l(\mu,\epsilon) &= \omega_{L+2}^{l-1}(\mu,\epsilon)\,, \\
\delta_{L+2}^l(\lambda)\omega_{L+2}^l(\mu,\epsilon) &= \frac{h(\lambda-\mu-\epsilon)}{h(\lambda-\mu-\epsilon+\eta)}\omega_{L+2}^{l+1}(\mu,\epsilon)\,, \\
\gamma_{L+2}^l(\lambda)\omega_{L+2}^l(\mu,\epsilon) &= 0\,.
\end{aligned}
\tag{75}
$$

From the local actions (73) and (75) it is straightforward, that the global state written as

$$
\Omega_l^+ = \omega_1^l \otimes \omega_2^l \otimes \cdots \otimes \omega_L^l \otimes \omega_{L+1}^l(\mu) \otimes \omega_{L+2}^l(\mu,\epsilon)\,,
\tag{76}
$$

is a proper reference state, satisfying the following relations:

$$
\begin{aligned}
A_l^+(\lambda)\Omega_l^+ &= \frac{h(\lambda-\mu)}{h(\lambda-\mu-\eta)}\Omega_{l-1}^+\,, \\
D_l^+(\lambda)\Omega_l^+ &= \frac{h(\lambda-\mu-\epsilon)}{h(\lambda-\mu-\epsilon+\eta)}\left(\frac{h(\lambda)}{h(\lambda+\eta)}\right)^L \Omega_{l+1}^+\,, \\
C_l^+(\lambda)\Omega_l^+ &= 0\,.
\end{aligned}
\tag{77}
$$

In order to obtain the eigenstates and eigenvalues of the enlarged transfer matrix, we again consider the more general transformations

$$
T_a^{+,k,l}(\lambda) = M_k^{-1}(\lambda)T_a^+(\lambda)M_l(\lambda) = \begin{pmatrix} A_{k,l}^+(\lambda) & B_{k,l}^+(\lambda) \\ C_{k,l}^+(\lambda) & D_{k,l}^+(\lambda) \end{pmatrix}\,.
\tag{78}
$$

The $R$-matrices introduced on the two extra sites satisfy the Yang-Baxter equation, therefore the $RTT$-relation (26) remains valid even for the enlarged monodromy matrix $T_a^+(\lambda)$. Since the transformation matrices are unchanged, the elements of the enlarged and gauged transformed monodromy matrix $T_a^{+(k,l)}(\lambda)$ satisfy the same commutation relations (45) as the operators $A_{k,l}(\lambda), B_{k,l}(\lambda), C_{k,l}(\lambda)$ and $D_{k,l}(\lambda)$. Therefore the eigenstates and eigenvalues of the enlarged transfer matrix can be obtained in the same way as for the original chain. The only differences compared to the formulae (49),(51) and (52) are that the eigenvalues of the operators $A_l(\lambda)$ and $D_l(\lambda)$ have to be replaced, in accordance with (77), and that the number of rapidities $n$ is increased by 1 (since the length of the chain is increased by 2). Accordingly, the state

$$
\Psi_\theta^+(\lambda_1,\ldots,\lambda_m) = \sum_{l=-\infty}^{\infty} e^{2\pi i l\theta}\Psi_l^+(\lambda_1,\ldots,\lambda_m)\,,
\tag{79}
$$

with

$$
\Psi_l^+(\lambda_1,\ldots,\lambda_m) = B_{l+1,l-1}^+(\lambda_1)\ldots B_{l+m,l-m}^+(\lambda_m)\Omega_{l-m}^+\,,
\tag{80}
$$

and $m = n+1 = L/2+1$, is an eigenstate of the enlarged transfer matrix, with eigenvalue

$$
\Lambda^+(\lambda|\mu,\epsilon) = e^{2\pi i\theta}\frac{h(\lambda-\mu)}{h(\lambda-\mu-\eta)}\prod_{k=1}^m \alpha(\lambda,\lambda_k) + e^{-2\pi i\theta}\frac{h(\lambda-\mu-\epsilon)}{h(\lambda-\mu-\epsilon+\eta)}\left(\frac{h(\lambda)}{h(\lambda+\eta)}\right)^L \prod_{k=1}^m \alpha(\lambda_k,\lambda)\,,
\tag{81}
$$

given that the modified Bethe equations are satisfied:

$$
\frac{h(\lambda_j-\mu-\eta)h(\lambda_j-\mu-\epsilon)}{h(\lambda_j-\mu)h(\lambda_j-\mu-\epsilon+\eta)}\left(\frac{h(\lambda_j)}{h(\lambda_j+\eta)}\right)^L = e^{4\pi i\theta}\prod_{\substack{k=1 \\ k\neq j}}^m \frac{\alpha(\lambda_j,\lambda_k)}{\alpha(\lambda_k,\lambda_j)}\,.
\tag{82}
$$

Once the enlarged transfer matrix eigenvalue (81) is obtained, the remaining task is to evaluate the differentiation, present in (70). To do this, we have to examine how the Bethe roots change

in the limit $\epsilon \to 0$. As it was shown in [20], at the limit $\epsilon \to 0$ the two extra sites decouple and (some of) the eigenstates of the enlarged chain take the form

$$t^+(\lambda)\big|\Psi^+\big\rangle = \Lambda_f(\lambda|\lambda_1,\ldots,\lambda_n)\Big(|\delta\rangle \otimes |\Psi(\lambda_1,\ldots,\lambda_n)\rangle\Big), \tag{83}$$

where $|\Psi(\lambda_1,\ldots,\lambda_n)\rangle$ is the eigenstate and $\Lambda_f(\lambda|\lambda_1,\ldots,\lambda_n)$ is the eigenvalue of the original chain. In this limit, the $m = n + 1$ rapidities go to the set $\{\mu, \lambda_1,\ldots,\lambda_n\}$, where $\{\lambda_1,\ldots,\lambda_n\}$ are the Bethe roots of the original chain. This can be seen by considering the transfer matrix eigenvalue (81):

$$e^{2\pi i\theta}\frac{h(\lambda-\mu)}{h(\lambda-\mu-\eta)}\alpha(\lambda,\tilde{\mu})\prod_{k=1}^{n}\alpha(\lambda,\tilde{\lambda}_k) + e^{-2\pi i\theta}\frac{h(\lambda-\mu-\epsilon)}{h(\lambda-\mu-\epsilon+\eta)}\left(\frac{h(\lambda)}{h(\lambda+\eta)}\right)^L\alpha(\tilde{\mu},\lambda)\prod_{k=1}^{n}\alpha(\tilde{\lambda}_k,\lambda), \tag{84}$$

which clearly goes to $\Lambda_f(\lambda|\lambda_1,\ldots,\lambda_n)$ in the limit $\epsilon \to 0$, $\tilde{\mu} \to \mu$, and $\tilde{\lambda}_i \to \lambda_i$ (using the fact that $h(\lambda)$ is odd).

Therefore in the limit $\epsilon \to 0$, the solutions of the Bethe equations (82) take the form

$$\tilde{\mu} = \mu + \epsilon\gamma + \mathcal{O}(\epsilon^2), \qquad \tilde{\lambda}_j = \lambda_j + \epsilon\Delta\lambda_j + \mathcal{O}(\epsilon^2). \tag{85}$$

Our numerical results suggest, that the parameter $\theta$ in the Bethe equations does not change as $\epsilon$ varies. Therefore, it is indeed enough to examine the $\epsilon$-dependence of the Bethe roots, and $\theta$ can be considered as a constant throughout the calculation.

The factor $\gamma$ can be obtained from the Bethe equation for the rapidity $\tilde{\mu}$:

$$\frac{h(\tilde{\mu}-\mu-\eta)h(\tilde{\mu}-\mu-\epsilon)}{h(\tilde{\mu}-\mu)h(\tilde{\mu}-\mu-\epsilon+\eta)}\left(\frac{h(\tilde{\mu})}{h(\tilde{\mu}+\eta)}\right)^L\prod_{k=1}^{n}\frac{\alpha(\tilde{\lambda}_k,\tilde{\mu})}{\alpha(\tilde{\mu},\tilde{\lambda}_k)} = e^{4\pi i\theta}. \tag{86}$$

Since for $|x| \ll 1$ the function $h(x)$ behaves linearly ($h(x) \approx x$), the $\epsilon \to 0$ limit of (86) leads to:

$$\frac{1-\gamma}{\gamma}\left(\frac{h(\mu)}{h(\mu+\eta)}\right)^L\prod_{k=1}^{n}\frac{\alpha(\lambda_k,\mu)}{\alpha(\mu,\lambda_k)} = e^{4\pi i\theta}. \tag{87}$$

Therefore $\gamma$ is given by:

$$\gamma = \frac{\mathfrak{a}(\mu)}{1+\mathfrak{a}(\mu)}, \qquad \text{with} \qquad \mathfrak{a}(u) = e^{-4i\pi\theta}\left(\frac{h(u)}{h(u+\eta)}\right)^L\prod_{k=1}^{n}\frac{\alpha(\lambda_k,u)}{\alpha(u,\lambda_k)}. \tag{88}$$

Similarly, $\Delta\lambda_j$ can be obtained from the logarithmic form of the other Bethe equations:

$$p(\tilde{\lambda}_j-\mu-\epsilon) + p(\tilde{\lambda}_j-\mu-\eta) + \delta(\tilde{\lambda}_j-\tilde{\mu}) + Lp(\tilde{\lambda}_j) + \sum_{k\neq j}\delta(\tilde{\lambda}_j-\tilde{\lambda}_k) = 2\pi\big(Z_j+2\theta\big), \tag{89}$$

where $Z_j \in \mathbb{Z}$, and the functions $p(\lambda)$ and $\delta(\lambda)$ are defined as:

$$e^{ip(\lambda)} = \frac{h(\lambda)}{h(\lambda+\eta)}, \qquad e^{i\delta(\lambda)} = \frac{h(\lambda+\eta)}{h(\lambda-\eta)}. \tag{90}$$

From (89), the vector of rapidity shifts $\Delta\lambda$ can be expressed as:

$$G \cdot \Delta\lambda = \mathbf{H}(\mu), \tag{91}$$

where

$$G_{jk} = \frac{\partial(2\pi Z_k)}{\partial\lambda_j}\bigg|_{\epsilon=0} = \delta_{jk}\left[Lp'(\lambda_j) + \sum_{l=1}^{n}\varphi(\lambda_j-\lambda_l)\right] - \varphi(\lambda_j-\lambda_k), \tag{92}$$

with $\varphi(\lambda) = \delta'(\lambda)$ is the Gaudin matrix. Important to note that the first three terms in (89) do not contribute to the Gaudin matrix, since their $\lambda$-derivatives vanish when $\epsilon$ is set to zero. The quantity $\mathbf{H}(\mu)$ is a column vector with entries given by:

$$H_k(\mu) = p'(\lambda_k - \mu) + \frac{\mathfrak{a}(\mu)}{1 + \mathfrak{a}(\mu)} \varphi(\lambda_k - \mu). \tag{93}$$

The derivative of the transfer matrix eigenvalue of the enlarged chain (81) then can be calculated by using (88) and (91). A somewhat lengthy and technical computation yields the result:

$$i\left( \frac{d}{d\epsilon} \log \Lambda^+(\nu|\mu, \epsilon) \Big|_{\epsilon=0} \right) = \mathbf{H}(\nu) \cdot G^{-1} \cdot \mathbf{H}(\mu) + l(\mu, \nu) + l(\nu, \mu), \tag{94}$$

with

$$l(\mu, \nu) = \frac{p'(\nu - \mu)}{(1 + \mathfrak{a}(\mu))(1 + \mathfrak{a}^{-1}(\nu))}. \tag{95}$$

To obtain the current mean values, one has to take the derivative of (94) with respect to $\nu$ and Taylor-expand it in $\mu$ and $\nu$ around zero. Since for $|u| \ll 1$ the function $\mathfrak{a}(u)$ behaves as $\mathfrak{a}(u) \sim u^L$, therefore we can safely substitute it with 0. As a result, $\mathbf{H}(\mu)$ can be approximated by $H_k(\mu) \approx p'(\lambda_k - u)$. Moreover, $l(\mu, \nu)$ can be simply taken to be zero. All these substitutions lead to the exact result for the current mean values:

$$\langle \lambda_1, \ldots, \lambda_n | \hat{J}^{\beta}_{\alpha}(j) | \lambda_1, \ldots, \lambda_n \rangle = \mathbf{q}'_{\beta} \cdot G^{-1} \cdot \mathbf{q}_{\alpha}, \tag{96}$$

where $\mathbf{q}_{\alpha}$ and $\mathbf{q}'_{\beta}$ are column vectors with elements given by

$$\left( \mathbf{q}_{\alpha} \right)_j = q_{\alpha}(\lambda_j), \qquad \left( \mathbf{q}'_{\beta} \right)_j = \frac{dq_{\beta}(\lambda_j)}{d\lambda}, \tag{97}$$

with $q_{\alpha}(\lambda)$ being the charge eigenfunctions, given by (55). We also used that $p'(u) = (-i)e(u)$ is the energy eigenfunction. With this, the proof of the main result (20) is complete.

In the Appendix C we also present numerical checks of the main result.

## 5 Conclusion and outlook

In this work, we derived the exact finite volume mean values of the current and generalized current operators in the XYZ model. The functional form of the final result is identical to that of the earlier results for the XXZ chains. This is a perhaps surprising finding, given that the XYZ model lacks $U(1)$-symmetry, therefore the structure of the Bethe eigenstates is very different from those of the XXZ models. A common property of the two classes of models is that the charge mean values can be expressed using the set of rapidities, which are determined by the Bethe equations. Our present results show that it is very natural and convenient to express also the current mean values using this set of dynamical variables.

The methods of this work go back to the classical paper [58], and they can be applied only in even volumes. It would be interesting to consider the current mean values also with other methods, perhaps via Separation of Variables (SoV).

In this work, we considered only the finite volume situation, the thermodynamic limit is left to further work. It would be important to compare the resulting formulas with those of earlier work, for example [63, 64]. This would help to understand the general structure of correlation functions in these models.

Finally, it would be interesting to work out Generalized Hydrodynamics for the XYZ spin chain. This could lead to exact predictions for the transport of conserved charges (for example

the energy, see [22]). The physical meaning of the rapidity variables is less clear in this model. It would be interesting to consider quantum quenches and also entanglement production in this model, and to compare with earlier results, which were derived for models with local $U(1)$-symmetries (see for example [73]). Our results for the current mean values are a first step in this direction.

## Acknowledgements

We are thankful to Tamás Gombor for very useful discussions.

## A   Elliptic functions

Here we present the definitions of the elliptic theta functions, and a list of their properties which are used in the main text. For more details on them see [74]. The theta functions can be defined as infinite sums:

$$
\begin{aligned}
\vartheta_1(u,q) &= -i\sum_{n=-\infty}^{\infty}(-1)^n q^{(n+1/2)^2}e^{i(2n+1)u} = 2\sum_{n=1}^{\infty}(-1)^{n+1}q^{(n-1/2)^2}\sin\big((2n-1)u\big),\\
\vartheta_2(u,q) &= \sum_{n=-\infty}^{\infty}q^{(n+1/2)^2}e^{i(2n+1)u} = 2\sum_{n=1}^{\infty}q^{(n-1/2)^2}\cos\big((2n-1)u\big),\\
\vartheta_3(u,q) &= \sum_{n=-\infty}^{\infty}q^{n^2}e^{i2nu} = 1+2\sum_{n=1}^{\infty}q^{n^2}\cos\big(2nu\big),\\
\vartheta_4(u,q) &= \sum_{n=-\infty}^{\infty}(-1)^n q^{n^2}e^{i2nu} = 1+2\sum_{n=1}^{\infty}(-1)^n q^{n^2}\cos\big(2nu\big).
\end{aligned}
\tag{A.1}
$$

Here q is the nome of the functions ($|q|<1$). The notation $\vartheta_j(u|\tau)$, ($j\in\{1,2,3,4\}$) is also used, where $\tau$ ($\operatorname{Im}\tau>0$) is called the parameter of the function, and is related to $q$ as $q=e^{i\pi\tau}$. For brevity we do not denote either $q$ or $\tau$, and simply use the notation $\vartheta_j(u)$. From (A.1) it is obvious that $\vartheta_1(u)$ is odd, while $\vartheta_2(u)$, $\vartheta_3(u)$ and $\vartheta_4(u)$ are even functions of $u$. The theta functions are quasiperiodic functions with periods $\pi$ and $\pi\tau$:

$$
\begin{aligned}
\vartheta_1(u+\pi) &= -\vartheta_1(u), & \vartheta_1(u+\pi\tau) &= -\frac{1}{q}e^{-2iu}\vartheta_1(u),\\
\vartheta_2(u+\pi) &= -\vartheta_2(u), & \vartheta_2(u+\pi\tau) &= \frac{1}{q}e^{-2iu}\vartheta_2(u),\\
\vartheta_3(u+\pi) &= \vartheta_3(u), & \vartheta_3(u+\pi\tau) &= \frac{1}{q}e^{-2iu}\vartheta_3(u),\\
\vartheta_4(u+\pi) &= \vartheta_4(u), & \vartheta_4(u+\pi\tau) &= -\frac{1}{q}e^{-2iu}\vartheta_4(u).
\end{aligned}
\tag{A.2}
$$

The theta functions also satisfy addition theorems, which are used throughout the main text:

$$
\begin{aligned}
\vartheta_1(u)\vartheta_1(v)\vartheta_1(w)\vartheta_1(u+v+w) &+ \vartheta_4(u)\vartheta_4(v)\vartheta_4(w)\vartheta_4(u+v+w)\\
&= \vartheta_4(0)\vartheta_4(u+v)\vartheta_4(u+w)\vartheta_4(v+w),\\
\vartheta_1(u)\vartheta_1(v)\vartheta_4(w)\vartheta_4(u+v+w) &+ \vartheta_4(u)\vartheta_4(v)\vartheta_1(w)\vartheta_1(u+v+w)\\
&= \vartheta_4(0)\vartheta_4(u+v)\vartheta_1(u+w)\vartheta_1(v+w),\\
\vartheta_4(u-v)\vartheta_1(u+v) - \vartheta_4(u+v)\vartheta_1(u-v) &= \frac{2\vartheta_1(v)\vartheta_2(u)\vartheta_3(u)\vartheta_4(v)}{\vartheta_2(0)\vartheta_3(0)}.
\end{aligned}
\tag{A.3}
$$

# B  Charge and current operators

Here we present the explicit form of the first few charge and current operators in the XYZ model. The set of local conserved charges is encoded by the transfer matrix. However, a more convenient way of explicitly calculating these charge operators is provided by the boost operator $\mathcal{B}$, which is defined on an infinite chain by the formal expression [75, 76]

$$\mathcal{B} = \sum_{j=-\infty}^{\infty} j h_{j,j+1}. \tag{B.1}$$

With the help of $\mathcal{B}$, the charge operators can be obtained recursively:

$$\hat{Q}_{\alpha+1} = i\big[\mathcal{B}, \hat{Q}_\alpha\big] + \text{const}. \tag{B.2}$$

The constant term is not fixed by the boost operator. In the expressions below, we chose them to match the construction coming from the transfer matrix (29). It is convenient to introduce the vectors $\underline{\sigma}_j$, $\underline{\hat{\sigma}}_j$ and $\underline{\tilde{\sigma}}_j$ with elements given by

$$\big(\underline{\sigma}_j\big)_a = \sigma_j^a, \qquad \big(\underline{\hat{\sigma}}_j\big)_a = \sqrt{J_a}\,\sigma_j^a, \qquad \big(\underline{\tilde{\sigma}}_j\big)_a = \sqrt{\frac{J_x J_y J_z}{J_a}}\,\sigma_j^a, \tag{B.3}$$

where $\sigma_j^a$ ($a = x, y, z$) is the appropriate Pauli matrix at site $j$ and the $J_a$'s are the coefficients in the Hamiltonian (1). Using this notation, the first three higher charge density can be written as [70]:

$$
\begin{aligned}
\hat{q}_3(j) &= -\frac{1}{2}\big(\underline{\hat{\sigma}}_j \times \underline{\tilde{\sigma}}_{j+1}\big) \cdot \underline{\hat{\sigma}}_{j+2}, \\
\hat{q}_4(j) &= \Big(\big(\underline{\hat{\sigma}}_j \times \underline{\tilde{\sigma}}_{j+1}\big) \times \underline{\tilde{\sigma}}_{j+2}\Big) \cdot \underline{\hat{\sigma}}_{j+3} + J_x J_y J_z \underline{\sigma}_j \cdot \underline{\sigma}_{j+2} + \sum_{a=x,y,z} J_a^2 \hat{\sigma}_j^a \hat{\sigma}_{j+1}^a \\
&\quad - 2\big(J_x^2 + J_y^2 + J_z^2\big) h_{j,j+1} + C_4, \\
\hat{q}_5(j) &= -3\Bigg\{ \Big[\big(\big(\underline{\hat{\sigma}}_j \times \underline{\tilde{\sigma}}_{j+1}\big) \times \underline{\tilde{\sigma}}_{j+2}\big) \times \underline{\tilde{\sigma}}_{j+3}\Big] \cdot \underline{\hat{\sigma}}_{j+4} + \sum_{a,b,c} \frac{J_x J_y J_z}{J_a} \epsilon_{abc}\big(\hat{\sigma}_j^a \tilde{\sigma}_{j+2}^b \hat{\sigma}_{j+3}^c \\
&\quad + \hat{\sigma}_j^b \tilde{\sigma}_{j+1}^c \hat{\sigma}_{j+3}^a\big) - \sum_{a,b,c} \epsilon_{abc} J_b^2 \hat{\sigma}_j^a \tilde{\sigma}_{j+1}^b \hat{\sigma}_{j+2}^c \Bigg\} - 4\big(J_x^2 + J_y^2 + J_z^2\big) \hat{q}_3(j).
\end{aligned}
\tag{B.4}
$$

Here $\cdot$ and $\times$ denote the usual scalar and vector products, respectively, while $\epsilon_{abc}$ is the Levi-Civita symbol. The constant $C_4$ is defined as:

$$C_4 = \frac{1}{2} \frac{d^3}{d\lambda^3}\Big(\log h(\lambda + \eta)\Big)\bigg|_{\lambda=0}. \tag{B.5}$$

The current operators describing the flow of the charges are defined through the continuity equation:

$$i[H, \hat{q}_\alpha(j)] = \hat{J}_\alpha(j) - \hat{J}_\alpha(j+1). \tag{B.6}$$

One can define generalized current operators as well, which describe the flow of the conserved charges under the time evolution governed by the charge $\hat{Q}_\beta$:

$$i[\hat{Q}_\beta, \hat{q}_\alpha(j)] = \hat{J}_\alpha^\beta(j) - \hat{J}_\alpha^\beta(j+1). \tag{B.7}$$

The first few of these current operators are given by:

$$\hat{J}_2(j) = \frac{1}{2}\left(\underline{\hat{\sigma}}_{j-1} \times \underline{\tilde{\sigma}}_j\right) \cdot \underline{\hat{\sigma}}_{j+1},$$

$$\hat{J}_3(j) = -\frac{1}{2}\left(\left(\underline{\hat{\sigma}}_{j-1} \times \underline{\tilde{\sigma}}_j\right) \times \underline{\tilde{\sigma}}_{j+1}\right) \cdot \underline{\hat{\sigma}}_{j+2} + \frac{1}{2}\left(\sum_{a=x,y,z} J_a^2 \hat{\sigma}_j^a \hat{\sigma}_{j+1}^a - 2\left(J_x^2 + J_y^2 + J_z^2\right)h_{j,j+1}\right),$$

$$\hat{J}_2^3(j) = -\frac{1}{2}\left\{\left(\left(\underline{\hat{\sigma}}_{j-2} \times \underline{\tilde{\sigma}}_{j-1}\right) \times \underline{\tilde{\sigma}}_j\right) \cdot \underline{\hat{\sigma}}_{j+1} + \left(\left(\underline{\hat{\sigma}}_{j-1} \times \underline{\tilde{\sigma}}_j\right) \times \underline{\tilde{\sigma}}_{j+1}\right) \cdot \underline{\hat{\sigma}}_{j+2}\right.$$

$$\left. + \sum_{a=x,y,z} J_a^2\left(\hat{\sigma}_{j-1}^a \hat{\sigma}_j^a + \hat{\sigma}_j^a \hat{\sigma}_{j+1}^a\right) - 2\left(J_x^2 + J_y^2 + J_z^2\right)(h_{j-1,j} + h_{j,j+1}) + 2J_x J_y J_z \underline{\sigma}_{j-1} \cdot \underline{\sigma}_{j+1}\right\}.$$

$$(B.8)$$

From (B.4) and (B.8) it can be seen, that $\hat{J}_2(j) = -\hat{q}_3(j-1)$.

## C  Numerical results

In order to check our main result (20), we numerically compared it to exact diagonalization. Numerical solutions to the Bethe equations for the XYZ model were considered previously in [59], in the framework of the so called Off-Diagonal Bethe Ansatz. However, to the best of our knowledge, in that paper the explicit values of the Bethe roots were only given for a special choice of $\eta$, which satisfies the constraint (56). Our result works for that special choice as well, but here we consider the general case. Therefore, we solved the Bethe equations (52) numerically (with $\theta = 0$), for randomly chosen parameters of the model, and calculated the charge and current mean values according to the generalized Bethe Ansatz solution and our main result. These results agreed with the ones obtained from the exact diagonalization with high numerical precision. Important to note, that the current operators (B.8) defined by the continuity equation only up to a free constant term. However, the current generating function (62) (and consequently our main result) is well-defined. As a result, the current mean values obtained from exact diagonalization may differ by a constant from the ones calculated by (20). We disregarded these constant terms, and in the tables below, displayed the results calculated according to (20). Unfortunately, solving the Bethe equations numerically for the whole spectrum proved to be a difficult task, and we were only able to obtain the Bethe roots for a subset of the eigenstates of the Hamiltonian. Nevertheless, in the cases found, the current mean values agreed with our main result.

In Table 1-4, we present the numerical results for chain lengths $L = 4$ and $L = 6$ with different parameters of the Hamiltonian. Concrete formulas for the real space representation of the charges and currents are found in the Appendix B.

Table 1: Bethe roots for $\eta = 0.922$, $q = 2.70 \times 10^{-3}$ and $L = 4$.

|  | $\lambda_1$ | $\lambda_2$ |
|---|---|---|
| 1. | -0.461 + 2.693 $i$ | -0.461 - 2.693 $i$ |
| 2. | -0.461 | 1.11 |
| 3. | 1.11 - 2.597 $i$ | -0.461 + 2.597 $i$ |
| 4. | -0.461 - 2.597 $i$ | 1.11 + 2.597 $i$ |
| 5. | 0.2075 - 1.479 $i$ | -1.129 + 1.479 $i$ |
| 6. | 1.11 + 1.479 $i$ | -0.461 - 1.479 $i$ |
| 7. | 1.11 - 2.406 $i$ | 1.11 + 2.406 $i$ |

Table 2: Charge and current mean values for $\eta = 0.922$, $q = 2.70 \times 10^{-3}$ and $L = 4$.

|   | $E$ | $\langle \hat{Q}_3 \rangle$ | $\langle \hat{Q}_4 \rangle$ | $\langle \hat{J}_2 \rangle$ | $\langle \hat{J}_3 \rangle$ | $\langle \hat{J}_2^3 \rangle$ |
|---|---|---|---|---|---|---|
| 1. | -5.952 | 0 | 0 | 0 | 7.499 | 0 |
| 2. | -3.034 | 0 | 38.224 | 0 | 0 | -9.556 |
| 3. | -1.538 | 6.277 | -9.391 | -1.569 | 2.348 | 2.348 |
| 4. | -1.538 | -6.277 | -9.391 | 1.569 | 2.348 | 2.348 |
| 5. | -0.116 | 0 | 0 | 0 | -0.144 | 0 |
| 6. | -0.041 | 0 | -0.661 | 0 | 0 | 0.165 |
| 7. | 1.456 | 0 | 0 | 0 | -0.312 | 0 |

Table 3: Bethe roots for $\eta = 0.620$, $q = 3.80 \times 10^{-4}$ and $L = 6$.

|   | $\lambda_1$ | $\lambda_2$ | $\lambda_3$ |
|---|---|---|---|
| 1. | -0.3102 + 0.1242 $i$ | 1.261 + 0.2323 $i$ | -0.3102 - 0.3565 $i$ |
| 2. | -0.3102 | -1.027 - 1.969 $i$ | 0.4069 + 1.969 $i$ |
| 3. | -0.3102 - 1.969 $i$ | -0.3102 | 1.261 + 1.969 $i$ |
| 4. | 1.261 + 0.6368 $i$ | 1.261 - 0.6368 $i$ | -0.3102 |
| 5. | 0.4093 - 2.063 $i$ | -1.03 + 1.875 $i$ | -0.3102 + 0.1875 $i$ |
| 6. | -1.03 - 1.875 $i$ | -0.3102 + 3.75 $i$ | 0.4093 - 1.875 $i$ |
| 7. | -3.452 + 4.125 $i$ | -3.452 - 2.06 $i$ | 1.261 - 2.065 $i$ |
| 8. | -0.3102 + 2.06 $i$ | 1.261 - 1.873 $i$ | -0.3102 - 0.1872 $i$ |
| 9. | -0.3102 - 0.1711 $i$ | 1.261 - 0.5394 $i$ | 1.261 + 0.7105 $i$ |
| 10. | 1.261 + 3.227 $i$ | -0.3102 - 3.767 $i$ | 1.261 + 0.5394 $i$ |
| 11. | 1.261 - 0.5463 $i$ | -0.6228 + 0.2731 $i$ | 0.002408 + 0.2731 $i$ |
| 12. | -0.6228 - 0.2731 $i$ | 0.002408 - 0.2731 $i$ | 1.261 + 0.5463 $i$ |
| 13. | -0.3102 + 0.6256 $i$ | 0.4476 + 1.656 $i$ | -1.068 - 2.282 $i$ |
| 14. | -1.068 - 1.656 $i$ | 0.4476 + 2.282 $i$ | -0.3102 - 0.6256 $i$ |
| 15. | -0.3102 - 0.6307 $i$ | -0.3102 - 1.634 $i$ | 1.261 + 2.265 $i$ |
| 16. | 1.261 + 1.673 $i$ | -0.3102 - 2.304 $i$ | -0.3102 + 0.6307 $i$ |
| 17. | -0.3102 + 0.4694 $i$ | 1.261 + 3.133 $i$ | 1.261 - 3.603 $i$ |
| 18. | 1.261 + 0.8047 $i$ | -0.3102 - 0.4694 $i$ | 1.261 - 0.3353 $i$ |
| 19. | 0.3602 + 1.969 $i$ | 1.261 | -0.9806 - 1.969 $i$ |
| 20. | -0.3102 - 1.969 $i$ | 1.261 | 1.261 + 1.969 $i$ |
| 21. | 1.261 - 0.939 $i$ | 1.261 - 2.999 $i$ | 1.261 + 3.938 $i$ |

Table 4: Charge and current mean values for $\eta = 0.620$, $q = 3.80 \times 10^{-4}$ and $L = 6$.

|  | $E$ | $\langle \hat{Q}_3 \rangle$ | $\langle \hat{Q}_4 \rangle$ | $\langle \hat{Q}_5 \rangle$ | $\langle \hat{J}_2 \rangle$ | $\langle \hat{J}_3 \rangle$ | $\langle \hat{J}_2^3 \rangle$ |
|---|---|---|---|---|---|---|---|
| 1. | -7.323 | -3.208 | 11.558 | 1024.882 | 0.535 | 10.821 | -1.926 |
| 2. | -6.260 | 0 | 133.886 | 0 | 0 | -0.022 | -22.314 |
| 3. | -6.246 | 0 | 133.802 | 0 | 0 | 0 | -22.300 |
| 4. | -5.385 | 0 | 133.987 | 0 | 0 | -0.214 | -22.331 |
| 5. | -4.535 | -13.460 | -5.165 | 571.976 | 2.243 | 6.685 | 0.861 |
| 6. | -4.535 | 13.460 | -5.165 | -571.976 | -2.243 | 6.685 | 0.861 |
| 7. | -4.528 | -13.468 | -5.054 | 574.289 | 2.245 | 6.678 | 0.842 |
| 8. | -4.528 | 13.468 | -5.054 | -574.289 | -2.245 | 6.678 | 0.842 |
| 9. | -3.870 | 13.460 | 5.165 | -686.283 | -2.243 | 5.727 | -0.861 |
| 10. | -3.870 | -13.460 | 5.165 | 686.283 | 2.243 | 5.727 | -0.861 |
| 11. | -1.741 | -3.154 | 8.965 | 134.472 | 0.526 | -0.400 | -1.494 |
| 12. | -1.741 | 3.154 | 8.965 | -134.472 | -0.526 | -0.400 | -1.494 |
| 13. | -1.085 | -3.189 | -11.482 | -48.155 | 0.531 | 1.584 | 1.914 |
| 14. | -1.085 | 3.189 | -11.482 | 48.155 | -0.531 | 1.584 | 1.914 |
| 15. | -1.083 | 3.208 | -11.558 | 48.281 | -0.535 | 1.591 | 1.926 |
| 16. | -1.083 | -3.208 | -11.558 | -48.281 | 0.535 | 1.591 | 1.926 |
| 17. | -0.848 | -5.843 | -22.634 | -88.228 | 0.974 | 2.491 | 3.772 |
| 18. | -0.848 | 5.843 | -22.634 | 88.228 | -0.974 | 2.491 | 3.772 |
| 19. | 0.607 | 0 | -1.487 | 0 | 0 | -0.020 | 0.248 |
| 20. | 0.637 | 0 | -1.499 | 0 | 0 | 0 | 0.250 |
| 21. | 1.214 | 0 | -0.524 | 0 | 0 | -0.132 | 0.087 |

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
