# Peer review of "Current mean values in the XYZ model"

_SciPost Physics, doi:SciPost Phys. 14, 158 (2023)_

## Round 1 · Referee Report · Anonymous (Referee 1) · 2022-12-5

Report

Yang-Baxter integrable quantum chains, like the XYZ model considered in the manuscript, possess a sequence of mutually commuting conserved charges that are generated by the logarithm of the transfer matrix of the underlying vertex model. The conserved charges are sums over local operators, the 'densities', of growing length of locality. Those operators satisfy continuity equations with corresponding `current densities'. The latter are central in recent attempts to find a hydrodynamic description of the properties of integrable quantum chains in the classical limit. In this context the expectation values of the current densities in excited states of the quantum chain are of high interest.

The authors obtain expressions for such expectation values in terms of Bethe roots. Along with the current densities they consider generalized current densities for which the Hamiltonian is replaced by one of the higher conserved charges as a generator of the time evolution. The main result of the paper is given in equation (20). This equation shows how the expectation value of the generalized current is determined by the charge eigenfunctions and by the Gaudin matrix.

The appealing feature of equation (20) is its great simplicity and the fact that is has the same structure as was previously obtained by the same authors for the XXZ model. This points to a much larger scope of validity of the result, also because the XYZ chain belongs to a more general class of integrable quantum chains that has no U(1) symmetry and for this reason defies a more simple-minded algebraic Bethe Ansatz solution.

Altogether the authors present an interesting and timely result that suits very well for publication in SciPost. The paper is presented in clear language and with sufficient detail. I recommend publication in the present form.

Requested changes

Here is a list of a few typos and small peculiarities I came across. I think in equation (7) the product of scattering phases should depend on the permutation. In line 2, page 7 it should be 'wave packets' instead of 'wave pockets'. In the second line under (25) it should be said 'isomorphic to' instead of 'equivalent to'. The notion 'computational basis elements' under equation (40) does not seem common to me. Better provide a definition. Above (83) '(some) of' should be '(some of)'. In the fifth line of section 5 on page 18 it should be 'two classes' instead of 'two class' and at the end of the third line on page 19 'spin chain' instead of 'spin chains'. I also recommend to go once more through the reference section before publication. There are issues with small and capital letters at several places, e.g. 'drude' instead of 'Drude' in [13]. For the reader's sake the authors might also consider to cite some of the classical papers in the appendices: The paper of M. G. Tetel'man, Sov. Phys. JETP 55 (1982) p 306 in appendix B and the works of J. D. Johnson and S. Krinsky and B. M. McCoy, Phys. Rev. A 8 (1973) p 2526 as well as of A. Klümper and J. Zittartz, Z. Phys. B 71 (1988) p 495 in appendix C.

---

## Round 1 · Referee Report · Anonymous (Referee 2) · 2023-1-11

Strengths

1- Exact result in an interacting integrable model 2- Generality of the approach

Weaknesses

1-Technical paper

Report

This paper provides an expression for the expectation value of current operators in the excited states of the XYZ model, which is an integrable model without a $U(1)$ symmetry. This supplements the previous (recent) findings, which, if we exclude noninteracting systems, are focussed on $U(1)$ symmetric integrable models. In the presence of interactions, the absence of a $U(1)$ symmetry is a big complication and the result of this paper stands out for its simplicity.
I think that the paper is well written and I strongly recommend its publication.

---

## Round 2 · List of Changes

We made almost all changes asked by the referee of the first report.
All changes were about typos, wrong use of english words, and one small mistake in a formula.
In Appendix B we added two additional references, one of them suggested by this referee.
On the other hand, this referee also suggested two more references for appendix C, but we did not add them. They are somewhat relevant in a broader sense, but they do not include finite size numerical data to compare to. This is why we did not add them. In any case this was just a suggestion from the referee.
All changes were about typos, wrong use of english words, and one small mistake in a formula.
In Appendix B we added two additional references, one of them suggested by this referee.
On the other hand, this referee also suggested two more references for appendix C, but we did not add them. They are somewhat relevant in a broader sense, but they do not include finite size numerical data to compare to. This is why we did not add them. In any case this was just a suggestion from the referee.

---

## Editorial Decision

published